# Physician-Customized Strategies for Reducing Outpatient Waiting Time in South Korea Using Queueing Theory and Probabilistic Metamodels

**DOI:** 10.3390/ijerph19042073

**Published:** 2022-02-12

**Authors:** Hanbit Lee, Eun Kyoung Choi, Kyung A. Min, Eunjeong Bae, Hooyun Lee, Jongsoo Lee

**Affiliations:** 1School of Mechanical Engineering, Yonsei University, Seoul 03722, Korea; rock_through@yonsei.ac.kr; 2College of Nursing and Mo-Im Kim Nursing Research Institute, Yonsei University, Seoul 03722, Korea; 3Severance Hospital, Yonsei University Healthcare System, Seoul 03722, Korea; kamin@yuhs.ac; 4College of Nursing and Brain Korea 21 FOUR Project, Yonsei University, Seoul 03722, Korea; benjandsy@gmail.com; 5Graduate School of Nursing, Yonsei University, Seoul 03722, Korea; solagratia417@gmail.com; 6Department of Nursing, College of Health and Welfare, Gangneung-Wonju National University, Wonju 25457, Korea

**Keywords:** operations research in health services, outpatient waiting time, queueing, statistical analysis, probabilistic meta-modeling

## Abstract

The time a patient spends waiting to be seen by a healthcare professional is an important determinant of patient satisfaction in outpatient care. Hence, it is crucial to identify parameters that affect the waiting time and optimize it accordingly. First, statistical analysis was used to validate the effective parameters. However, no parameters were found to have significant effects with respect to the entire outpatient department or to each department. Therefore, we studied the improvement of patient waiting times by analyzing and optimizing effective parameters for each physician. Queueing theory was used to calculate the probability that patients would wait for more than 30 min for a consultation session. Using this result, we built metamodels for each physician, formulated an effective method to optimize the problem, and found a solution to minimize waiting time using a non-dominated sorting genetic algorithm (NSGA-II). On average, we obtained a 30% decrease in the probability that patients would wait for a long period. This study shows the importance of customized improvement strategies for each physician.

## 1. Introduction

The most important factor in outpatient care is to provide an affordable service to a large number of patients [1]. However, outpatient departments (OPDs) are often overcrowded, and waiting times are long. The term waiting time refers to the time a patient spends waiting before being attended to by a healthcare professional in a hospital [2,3]. According to the Patient’s Charter of the United Kingdom Government, all patients must be seen within 30 min of their scheduled appointment time [4]. Waiting time is an important parameter in determining the quality of care and can be a valuable tool for evaluating patient satisfaction [2]. Previous studies have shown that a prolonged waiting time can be associated with low patient satisfaction [4,5,6,7]. Therefore, reducing waiting time has become a major issue not only for patient satisfaction but also for improving the quality of healthcare.

To reduce patient waiting time, efficient scheduling is required [8]. Scheduling focuses on procedures that determine how patient appointments (with healthcare professionals) are scheduled, both in terms of when and how they are set on a given day. More specifically, this involves rules that determine when appointments can be made (namely, morning versus afternoon) and the length of time (spacing) between appointments. This may also be extended to include designating the specific type of medical staff responsible for treating patients and the clinic space required to deliver the necessary treatments [8]. Efficient scheduling not only improves the satisfaction of patients and healthcare professionals, but also allows healthcare professionals to increase their efficiency in the treatment of more patients while reducing the waiting time [9].

Although we acknowledge that improving scheduling is not easy because it requires system-level transformation, such transformation can uncover previously unrecognized resources and improve all aspects of care delivery [9]. In previous studies, several interventions have been aimed at improving patient satisfaction by reducing waiting times in an OPD [10,11,12,13]. However, only a few have been well documented, their effects have rarely been evaluated with robust methods, and the OPD waiting time has not been a focus of the study. Therefore, to reduce long waiting times in an OPD, the associated parameters should be systematically analyzed based on real longitudinal data, and problem-solving methods should be designed using a theory-based approach.

From this perspective, queueing theory can provide an effective and powerful modeling technique that can help reduce waiting times [8]. The ultimate objective of queueing theory is to achieve an economic equilibrium between service cost and the time patients waste while waiting in a queue for a consultation [14]. Although several previous studies demonstrated that this theory can be used to design and implement an efficient scheduling system to improve the quality of hospital service [15,16,17,18,19], little research has been reported regarding the waiting time in tertiary hospital OPDs. Additionally, because most of these studies have been conducted in developed Western countries, it is necessary to identify whether the findings can be adapted to various healthcare systems in other countries.

Korea is in a unique position to observe whether patients choose a healthcare facility with an appropriate level of care when equal access to all levels of care is guaranteed by national health insurance [6]. This has been a controversial issue among patients, healthcare providers, and the Korean government for a long time [20]. Patients are claiming the right to choose the most qualified physicians, who, they believe, are in renowned hospitals that are typically located in large metropolitan areas [21]. Consequently, tertiary hospitals, which comprise only 3% of all hospitals, account for 27% of hospital outpatient expenditures for primary care patients [7]. Moreover, patients’ preference for a tertiary hospital is more than just a prejudice. A study on cancer treatment in Sweden showed higher survival rates in hospitals with higher operation volumes [22]. Hence, a major problem faced by hospitals when it comes to the functioning of the OPD is overcrowding and extended waiting periods [23].

The Korean government has attempted to address the problem of patients’ preference for tertiary hospitals with various measures; however, these have not proved to be effective [23]. In addition, decentralization efforts to improve patient concentration are not easily addressed at the hospital level; this is because the necessary increases in decision-making authority as well as the sustainability of financial and human resources and government policies have not been ensured [24]. Therefore, health services must rely on improved flow control and better capacity allocation to minimize the negative consequences of long waiting times. Simple penalty policies or competition between hospitals to reduce patient waiting times have not been useful. Instead, these temporary actions have increased waiting times [25]. Putting more money into the system may also lead to temporary improvements, but the results do not tend to last long [15,26]. Therefore, innovative organizational and structural changes must be introduced with purposeful planning and demand-oriented scheduling of outpatient care at the hospital level. If these innovative changes prove their effectiveness in reducing waiting times, they could be adopted by other countries as well.

This study aims to evaluate the parameters affecting long waiting times in OPDs based on real longitudinal data from a tertiary hospital and to optimize OPD schedules using queueing theory. Specifically, the aims are as follows:To analyze the actual outpatient waiting time data of a tertiary hospital;To apply queueing theory to the data of each physician to analyze the effect of influencing parameters on outpatient waiting time;To suggest ways to improve the waiting time in the OPD without the need for additional resources or reducing the number of patients.

The study flow is shown in Figure 1.

## 2. Theory and Data

### 2.1. Queueing Theory

Queueing theory helps map the queues that occur when patient demands are more than the throughput and enables the utilization of the system, average waiting time, arrival rate of patients, throughput of the system, and other changes to be calculated [27]. Queueing theory was initially developed to identify and improve the statistical characteristics of telephone switchboards. Ever since, it has been updated to apply to all systems with queues, such as restaurants, factories, and computerized data processing.

Models in queueing theory follow the notation proposed by Shortle et al. [27]. They are expressed as A/S/c/K/N/D, where A represents the probability distribution of patient arrivals—that is, the interval between the arrival times of the patients; S represents the probability distribution of the service time; c represents the number of physicians in the overall system; K represents the maximum capacity of the system, N represents the size of the patient pool; and D represents the service discipline. K, N, and D are often omitted; this happens when K and N are infinite and D is a first-in, first-out rule that ensures that the patients are served in the order they arrived.

In this study, we applied the M/M/1 model to the outpatient queueing model, as illustrated in Figure 2. It is assumed that the visit time intervals of the patients arriving at the hospital follow a Poisson distribution, and the physician’s consultation time follows an exponential distribution. For an outpatient session at the hospital, the analysis is performed using the queueing model for each session of the corresponding physician, as one physician cannot see the patient of another physician. Therefore, the number of servers is set to one. The system capacity and patient pool are assumed to be infinite, and the service discipline is first-in, first-out order. Queueing theory was used to calculate the probability that a patient will wait more than 30 min in each session, and then, the effectiveness of the parameters was analyzed. The method for calculating the probability that a patient waits for over 30 min is described in detail in the next section.

### 2.2. Outpatient Record Data

Most of the record data required for this study were taken from the data collected for outpatient waiting time management by the nursing division of Severance Hospital of Yonsei University Health System in the Republic of Korea. We used data from fifty-five physicians in three departments (two internal medicine clinics and one surgery clinic) from January 2016 to December 2017. The author who collected these data is a member of the nursing division of Severance Hospital, and their access to and utilization of the data collected by the nursing division was approved by the hospital.

Outpatient consultation at the hospital is divided into morning and afternoon sessions. Usually, one physician takes one session per day (in the morning or afternoon). In these data, this session is treated as the minimum unit, and the characteristic values of each session are recorded. The data include 613 consultation sessions, 71 of which are sessions that were missed, and hence, the actual number of sessions is 542. These data include records for 20.870 patients.

The data include the physician’s name and department, visit date, the start time of the consultation, the number of patients who waited for more than 30 min, whether a physician started the consultation late or early, the reasons for delays in the consultation, and the number of patients registered during that session. These data were managed by the nursing division and reviewed by the internal research committee of the division. Only the necessary anonymized data were extracted and provided to the research team.

To analyze the data from this study using queueing theory, additional data were required, such as the closing time of the outpatient clinic, the first and last appointment times, the number of first-time patients, the number of no-shows, and the number of patients who visited without an appointment. These additional data were collected retrospectively by a co-author authorized to access the outpatient electronic medical record system at Severance Hospital of Yonsei University Health System.

In this study, all the identifiable variables, including department, physician, and patient-level identification numbers, were anonymized to protect privacy. The study protocol was approved by the institutional review board (IRB) of Yonsei University Health System (IRB No. Y-2018-0103).

The waiting time of the OPD was expected to be affected by seven parameters: the session time of day (○), day of the week (●), session month (◇), start time delay (◆), number of walk-in patients (□), number of no-shows (■), and proportion of first-time patients (☆). Here, the session time of day is used to distinguish between morning and afternoon consultations, and the start time delay refers to the difference between the scheduled and actual consultation start times. If the actual consultation was started earlier than the scheduled time, the start time delay parameter has a negative value.

## 3. Data Processing and Problem Formulation

### 3.1. Data Pre-Processing and Parameter Extraction

The data were first pre-processed for the subsequent statistical and queueing theory analyses. After dividing the data into three parts according to medical department, they were further divided according to the day of the week and month of the session according to the date. The start time delay values were derived by subtracting the scheduled start time of consultation from the actual start time. Hence, if the consultation started earlier than the scheduled time, the delay in start time was negative. The proportion of first-time patients was derived by dividing the number of first-time patients by the total number of patients in the session. Finally, the proportion of patients waiting for more than 30 min was derived by dividing the number of patients waiting for more than 30 min by the total number of patients in a particular session. The emphasis on the proportion of first-time patients and over 30 min waiting time is because these two values should increase linearly with the increase in the total number of patients in each session. However, according to practitioners, the number of no-shows has a more direct effect on patient waiting in a session than the ratio of no-shows.

Queueing theory was used to calculate the probability that a patient will wait more than 30 min for each session. The probability that the patient will wait more than *x* hours is calculated as follows.
(1)$$\begin{array}{c} P\left(t>x\right)=\rho \cdot e^{-\frac{x}{\bar{T}}}\\ \bar{T}=\frac{1}{\mu \left(1-\rho \right)}=\frac{1}{\mu -\lambda } \end{array}$$

In Equation (1), *ρ* represents the traffic intensity, which is *λ*/*μ*, where *λ* represents the rate of patient arrival, that is, the number of patients arriving per hour, which is the total number of patients subtracted by 1 in a session divided by the appointment time. The appointment time is the duration between the session start time and session end time. Moreover, *μ* denotes the physician’s processing power, that is, the average number of patients a physician can see per hour. As this value cannot be directly calculated from the data, the estimated value *μ_est_*, which is calculated as follows, was used.
(2)$$\mu_{est}=2W\left(\frac{x\lambda e^{x\lambda }}{R_{est}}\right)$$

Equation (2) is the inverse of Equation (1). In this inverse function, *W* denotes the Lambert function, which is expressed as follows.
(3)$$xe^{x}=a\;↔x=W\left(a\right)$$

Parameter *R* refers to the proportion of patients waiting for more than 30 min. In Equation (2), *R_est_* has the same value as *R* unless *R* is 0, in which case, the value estimated using a 95% confidence interval is used. Parameter *R_est_* is calculated as follows.
(4)$$R_{est}=1-{\left(0.95\right)}^{1/N}$$

Here, *N* refers to the number of treated patients. We calculate *R_est_* in this manner because when its value is 0, *μ_est_* becomes infinite, which is not realistic. Finally, *P*, the probability of over-waiting when *x* equals 0.5, is calculated using *μ_est_* instead of *μ* in Equation (1). The calculated probability of over-waiting can be changed when adjusting the number of patients in a session, which helps calculate the expected number of patients who will wait for more than 30 min.

### 3.2. Statistical Analysis of Effective Parameters

Statistical analysis was used to evaluate the significance of the seven parameters for the entire OPD and each of the three departments within it using IBM SPSS Statistics 25.(IBM, New York, USA). Table 1 presents the significance of seven effective parameters using statistical verification methods. A *t* test, analysis of variance (ANOVA), and multiple regression analysis using multi-variable linear regression models were performed according to the data type of each effective parameter. In multiple regression analysis, the probability of over-waiting was set as the dependent variable, and lateness of start time, number of receipts on the day, number of no-shows, and proportion of first-timers were set as independent variables.

### 3.3. Regression Modeling of Waiting Time Reduction

The waiting time minimization for each physician excludes those with a proportion of over-waiting patients that is less than 2.5% or those with fewer than six sessions in the data to ensure statistical significance. Of the total 55 physicians included in the data, 21 had an over-waiting rate of more than 2.5% and had attended more than six sessions. This corresponds to 246 of the 542 session datasets. In this study, the probabilities of over-waiting were normalized with respect to the physician with the highest probability of over-waiting to better express the study results. Table 2 presents the number of datasets (sessions), total number of patients, and normalized probability of over-waiting patients for the 21 physicians studied for waiting time minimization.

A metamodel is required for each physician to minimize the over-waiting probability of his or her patients. Therefore, queueing theory was used to calculate the over-waiting probability for each session, and a metamodel was constructed for calculating the patient’s over-waiting probability for each physician. Figure 3 shows the metamodel constructed for the session time of day parameter of physician C-007, who had the highest over-waiting probability. The metamodel utilizes the least squares method in the form of an exponential function using base *e* (Euler’s number), as follows.
(5)$$y=ae^{bx}$$

The data points used to construct the metamodel were computed from the actual outpatient waiting data and the over-waiting probability using queueing theory. In the metamodel, if the total number of patients in a session is changed by varying a weighting variable, the probability of over-waiting can be calculated using queueing theory, and the result is multiplied by the total number of patients to determine the number of over-waiting patients expected for that particular session. Subsequently, the total number of over-waiting patients for all sessions of a physician is divided by the total number of patients who consulted that physician to determine the probability of over-waiting. The metamodel can hence be used to calculate the probability of over-waiting for each physician when the total number of patients changes. In Figure 3, the y-axis represents the probability of over-waiting, and the x-axis represents the weighting variable for adjusting the total number of patients. The metamodel also represents the over-waiting probability calculated from the actual data with a coefficient of determination *R*^2^ of 0.99 or more.

Figure 3 shows the metamodel for the morning and afternoon, which are the categories of the session time of day parameter. The probability of over-waiting was divided by the results of physician C-007. The metamodels for the remaining six parameters were also constructed, and their parameter values are listed in Table 3. Of the 20 parameter categories shown in the table, categories with zero patients were excluded from the optimization process. For example, physician C-007 did not perform outpatient consultations on Monday, Friday, and Saturday, and therefore, the optimization was performed for the categories of Tuesday, Wednesday, and Thursday. Additionally, physician C-007 did not see any walk-in patients; therefore, there is no optimization based on the number of walk-in patients.

### 3.4. Formulation of the Waiting Time Reduction Problem

Using the constructed metamodel, the weighting of patients was optimized according to the effective parameters. First, the optimization problem was formulated. The value that had to be computed was the weighting of the number of patients that is best for each parameter. The objective was to minimize the probability of over-waiting for each parameter by adjusting the weight for the number of patients in the range of 90–110%. In addition, constraint conditions were set so that the total number of patients remained consistent.
Find  *x_ij_* (*i =* 1 to 7, *j* = 1, 2, …)(6)
Minimize  *P_i_*(*x_ij_*) = *P*_1_(*x*_11_, *x*_12_), *P*_2_(*x*_21_, *x*_22_, *x*_23_), *P*_3_(*x*_31_, *x*_32_), …(7)
Subject to  0.9 ≤ *x_ij_* ≤ 1.1(8)

Total Patients Number (*x_ij_*) = constant;

*i*: effective parameters;

*j*: subcategories for each effective parameter.

In this problem, 20 variables *x_ij_* were used to represent the weighting of the patients according to each category of the effective parameters, 7 objective functions *P_i_*(*x_ij_*) were used to represent the probability of over-waiting according to the 7 parameters, and 1 constraint was used to keep the total number of patients constant. Here, *i* indicates the parameter, and *j* indicates the category of each parameter. For example, the session time of day parameter had morning and afternoon categories, whereas the day of the week parameter had Monday, Tuesday, Wednesday, Thursday, Friday, and Saturday as categories.

The present study explored two types of waiting time optimization: (1) single-parameter optimization and (2) multi-parameter optimization. In single-parameter optimization, the probability of over-waiting was minimized using each of the effective parameters for each of the 21 physicians, whereas multi-parameter optimization was performed using all seven effective parameters together.

For the optimization, we used a non-dominated sorting genetic algorithm called NSGA-II, which is a type of multi-objective evolutionary algorithm. NSGA-II uses the crowding distance method, which controls the distance between correct answers to obtain the most evenly distributed solutions within the user-specified variable range. Furthermore, it uses the elitist principle, which selects the answers closest to the objective of the previous generation when moving to the next generation to determine the closest solution. This algorithm also introduces constraints to determine the optimal solution that satisfies multiple constraints simultaneously [21]. In this algorithm, one can set the population size and number of generations, which in this study are 100 and 500, respectively. Hence, the objective of the aforementioned optimization problem is to determine the optimal values of the 20 weighting variables that correspond to the categories of the seven parameters for all 21 physicians.

## 4. Results

### 4.1. Validation of the Effective Parameters

The results show that several effective parameters have a statistically significant effect on the proportion of over-waiting patients; however, the goodness-of-fit (*R*^2^) of the statistical model was not found to be sufficiently high. Specifically, the result of multiple regression analysis of the entire OPD shows that the significance probability of the number of walk-in patients parameter is 0.001, which is significant at 0.01, but the R^2^ of the statistical model is 0.026, which is extremely low. The ANOVA results of department A reveal that the significance probability of the day of the week parameter is 0.029, which is significant at 0.05, but none of the days have a significance probability of 0.1 or less in the post hoc test. The results of multiple regression analysis of department A show that the significance probability of the number of walk-in patients parameter is 0.043, which is significant at 0.05, but the R^2^ of the statistical model is 0.026, which again is extremely low. The multiple regression analysis results of department B show that the significance probability of the number of no-shows parameter is 0.005, which is significant at 0.01, but the R^2^ of the statistical model is 0.059, which is also extremely low. The results of multiple regression analysis of department C show that the significance probabilities of the number of no-shows and proportion of first-time patients parameters are 0.009 and 0.016, respectively, which are significant at 0.01 and 0.05, but the R^2^ of the statistical model is low at 0.108.

The results of this statistical analysis show that the analysis and minimization of the effect on the waiting times of the entire OPD and each department do not have a valid meaning. Hence, an analysis was conducted for individual physicians instead of for each department.

### 4.2. Results of Waiting Time Reduction

Table 4 details the results of single-parameter optimization for physician C-007. The optimal number of weighting variables for each of the 20 categories of the seven parameters was derived to obtain the smallest possible over-waiting probability. During optimization, the total number of patients was kept constant at 401; that is, each patient in each category was multiplied by their corresponding weighting, and then, the resulting value was added to the effective parameters.

In the case of physician C-007, the adjustment of the number of patients according to the start time delay parameter showed the highest improvement in the probability of over-waiting, followed by the number of no-shows and session month parameters, which were revealed to be the main effective parameters. The single-parameter optimization results for representative examples of physicians C-007 and B-006 are shown in Table 4 and Table 5, respectively. The three most effective parameters for C-007 are (1) start time delay, (2) number of no-shows, and (3) session month in that order, whereas the three most effective parameters for physician B-006 are (1) session month, (2) number of no-shows, and (3) day of the week in that order. Thus, the benefit of single-parameter optimization is that it provides the most effective solution for each physician to reduce the probability of over-waiting patients as much as possible. Because various characteristics (e.g., the affiliated department or personal schedule) of each physician affect the consultation hours and waiting time, this study recommends physician-customized strategies for reducing outpatient waiting time.

The results of the multi-parameter optimization of all physicians are presented in Table 6 and Figure 4. The minimization results of the over-waiting probability were obtained after the multi-parameter optimization was performed. For physician C-007, the probability of over-waiting through multi-parameter optimization was reduced from 1.00 to 0.82, resulting in an 18% improvement. As noted in Table 6, the main effective parameters and the improvement values after optimization differed for each physician. The major parameters for most physicians were day of the week and session month. By contrast, the parameters least often selected as most significant were start time delay and proportion of first-time patients.

The improved patient waiting time results of the 21 physicians were combined with the data from the remaining 34 physicians excluded from the optimization process, and the improvement of the entire OPD and each department was evaluated. The results are presented in Table 7 and Figure 5. From the viewpoint of the entire department, the normalized value of the probability of over-waiting decreased by approximately 30% (from 0.232 to 0.162) and by 31%, 42%, and 25% for departments A, B, and C, respectively. In the case of department C, the optimized rate of decrease was relatively small; however, the existing probability of over-waiting was high at 0.513, and its improvement was also the largest (0.126). The most significant effect was expected in department C because the goal of the study was to maintain the over-waiting rate below a certain level.

On average, a 30% decrease in the probability of over-waiting was observed. Considering that there were no valid parameters when the significance of effective parameters was evaluated by department, this improvement is quite substantial. Once again, this shows the necessity and possibility of improvement strategies customized for each physician.

## 5. Discussion

According to the basic laws of queueing theory, the average number of patients in the entire system is equal to the product of the arrival rate of the patients and the average time that each patient spends on the system [27]. Consequently, patient waiting time will not reduce unless additional resources are dedicated to speeding up patient processing or limiting the total number of patients processed by the system. However, in this study, instead of optimizing the entire system all at once, physician-customized strategies were optimized to improve patient waiting times without reducing the total number of patients or investing in additional resources. This utilization of physician-customized strategies to reduce waiting times is unique to our study. Scheduling and appropriate healthcare services are complex issues that require balancing clinical criteria and acuity; patient needs; and organizational resources, structure, and culture [9]. According to a review by the Organization for Economic Co-operation and Development on the strategies of 13 countries to reduce waiting times, countries utilizing more resources to increase production generally reported no lasting effect [28]. Therefore, it is worth investigating queueing theory in the healthcare system if it has the potential to bring about the maximum improvement with a minimal investment in resources.

In this study, statistical analysis was used to analyze the effective parameters for the entire OPD; however, for each of the three departments in the OPD, significant parameters could not be identified. Therefore, no significant optimization could be performed at the department level. However, owing to the optimization of the strategy for each physician obtained using queueing theory, the average probability of over-waiting was reduced by 30%. In the case of department C, which had the highest over-waiting probability, the normalized probability of over-waiting reduced from 0.513 to 0.387 (an improvement of 0.126, which is the highest value in this study). In the case of department B, the improvement was 42% (a decrease from 0.170 to 0.099), thereby achieving the highest improvement rate. These improvements demonstrate the potential for physician-customized improvement strategies. It is therefore necessary to manage the patient waiting times via a physician-specific analysis and optimization strategy rather than an across-the-board hospital service policy such as limiting the number of patients that can be booked and beginning outpatient care earlier than usual. This approach is similar to that taken in personalized healthcare, which can be implemented by using currently available technologies and knowledge to provide a market for the rational introduction of new personalized medicine tools [29].

Korea provides universal health coverage to its citizens; thus, patients prefer going to a tertiary hospital to receive a higher level of healthcare. These patients include those with minor conditions [7]. The level of competition among hospitals and clinics in Korea is very high because of the undifferentiated roles and functions of each type of healthcare institution [23]. The economic role of outpatient clinics has been gaining importance because of the increasing number of patients in tertiary care hospitals [6]. Therefore, a major problem faced by hospitals in Korea is the efficient management of outpatient waiting times [23]. Although the use of convenient systems in large hospitals, such as convenient check-in and check-out processes as well as reservation systems, has increased patient satisfaction, among the five domains of patient satisfaction (i.e., physician services, nurse services, technician services, convenience, and physical environment of the facility), convenience was rated as the lowest [6]. Unfortunately, this cannot be resolved easily because the problem is already at an advanced stage, and solving it could cause further inconvenience in Korea. Thus, a new method is needed to shift the mindset of people and efficiently deploy limited resources. To address this problem, this study explored queueing theory as a solution. In other words, if we evaluate the waiting time of each department and each physician and apply an individualized strategy obtained using queueing theory to target the physicians with the longest waiting times, we can expect overall hospital-level waiting times to decrease.

The major limitations of this study are as follows. Because improvements can only be observed post-optimization and are based on the data of the consultation sessions, the actual improvements could be lower than the improvements observed in the models. In addition, because the data were obtained from 542 sessions, the statistical significance of the results is low because the volume of data is relatively small for each physician. Additional data acquisition and actual field application studies will therefore be required in future.

This study identified the parameters that have a major influence on patient waiting times for each physician. However, establishing a strategy for improvement based on these parameters can be challenging; furthermore, in some cases, improvements may be difficult to achieve given the characteristics of the effective parameters. Therefore, the level of waiting time improvements suggested in this study could be challenging to achieve. In future studies, statistically meaningful results should be derived by expanding the scope of the departments, and additional data for each physician must be obtained. Moreover, a verification study is required to confirm the practical levels of improvements obtained using the optimized effective parameters for each physician in actual outpatient consultations. Finally, an opportunity for further research would be to examine other qualitative parameters that affect patient waiting times.

## 6. Conclusions

Waiting time in outpatient care is an important parameter in determining the quality of care and can be a valuable tool for evaluating patient satisfaction. Reducing waiting time is not easy because it requires system-level transformation. The results of this study using queueing theory suggest that hospital-level transformation is possible. Optimizing the strategy of each physician according to queueing theory achieved a 30% reduction in the average probability of waiting time. This utilization of physician-based strategies to reduce the waiting time is unique to our study. Therefore, in a healthcare environment with limited hospital-level resources, achieving maximum improvement with minimal resource investment through queueing theory makes it worth investigating in the healthcare system.

## Figures and Tables

**Figure 1 ijerph-19-02073-f001:**
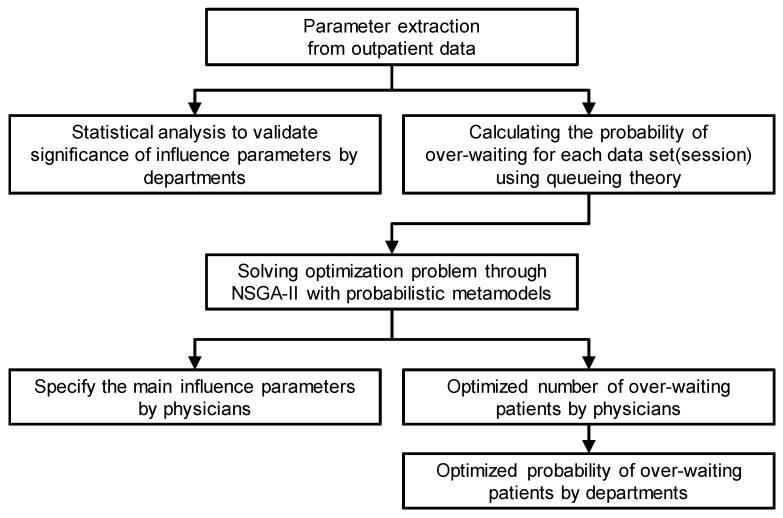
Overall research process of the present study.

**Figure 2 ijerph-19-02073-f002:**
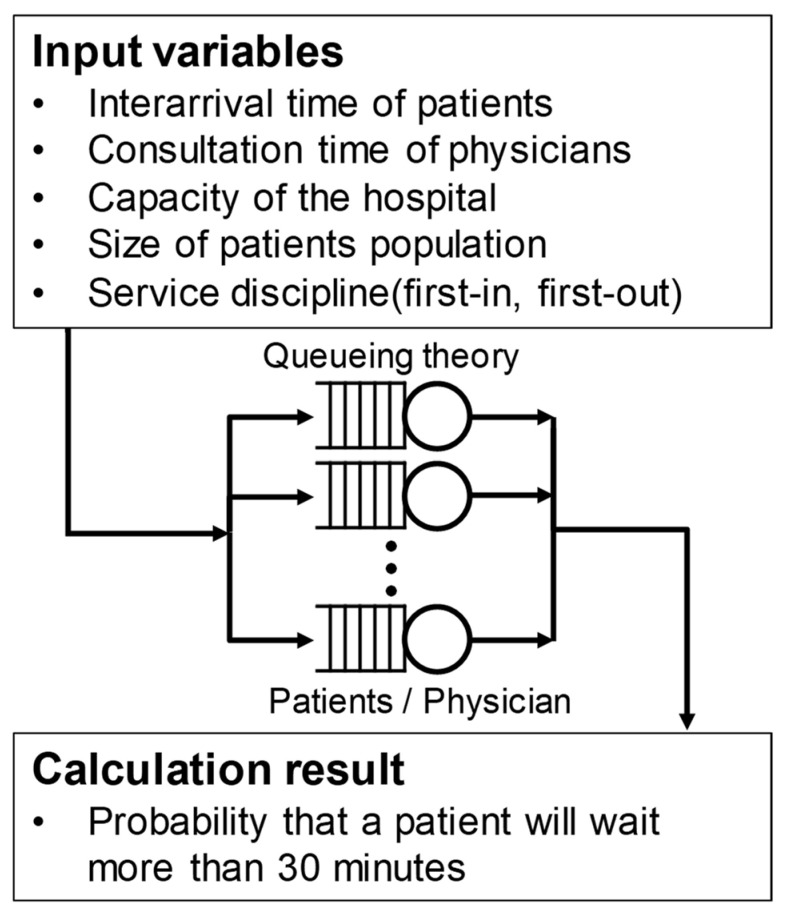
Schematic of queueing theory.

**Figure 3 ijerph-19-02073-f003:**
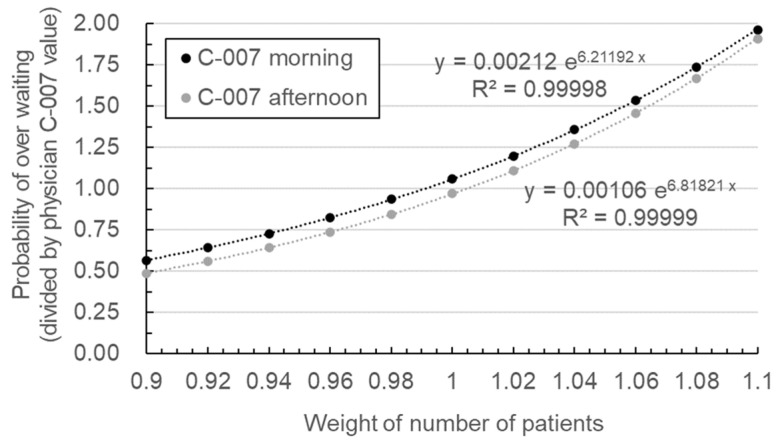
Metamodel of the probability of over-waiting patients for physician C-007.

**Figure 4 ijerph-19-02073-f004:**
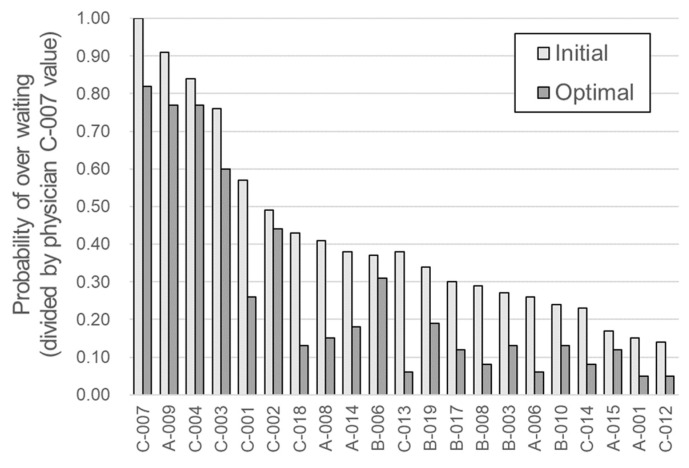
Optimization results for each of the 21 physicians.

**Figure 5 ijerph-19-02073-f005:**
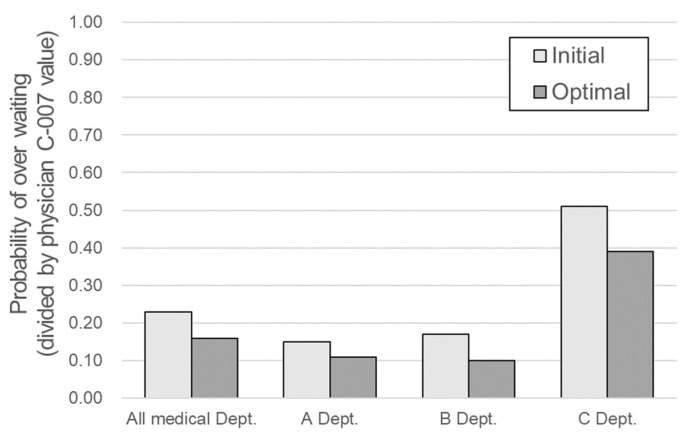
Optimization results for each of the medical departments.

**Table 1 ijerph-19-02073-t001:** Statistical analysis of significance.

Effective Parameters		Probability Statistics Verification Methods	All Three Medical Dept.	A Dept.	B Dept.	C Dept.
1. ○	Session running time	*t* test	X	X	X	X
2. ●	Day of the week	ANOVA	X	O	X	X
3. ◇	Session running month	ANOVA	X	X	X	X
4. ◆	Lateness of start time	Multiple Regression	X	X	X	X
5. □	Number of receipts on the day	Multiple Regression	O	O	X	X
6. ■	Number of no-shows	Multiple Regression	X	X	O	O
7. ☆	Proportion of first-timers	Multiple Regression	X	X	X	O

O: significant, X: insignificant.

**Table 2 ijerph-19-02073-t002:** Over-waiting outpatient data sorted by physician.

Physician ID Code	Number of Sessions	Total Number of Patients	Normalized Probability of Over-Waiting Patients
C-007	12	401	1.00
A-009	13	815	0.91
C-004	8	348	0.84
C-003	12	495	0.76
C-001	7	198	0.57
C-002	19	715	0.49
C-018	10	290	0.43
A-008	12	274	0.41
A-014	10	532	0.38
B-006	12	572	0.37
C-013	9	234	0.38
B-019	11	484	0.34
B-017	13	532	0.30
B-008	11	431	0.29
B-003	10	328	0.27
A-006	9	181	0.26
B-010	12	569	0.24
C-014	12	443	0.23
A-015	18	1033	0.17
A-001	14	807	0.15
C-012	12	390	0.14

**Table 3 ijerph-19-02073-t003:** Metamodel results for the probability of over-waiting patients for physician C-007.

Effective Parameters		Number of Patients	a	b
○	Morning	134	0.002121	6.212
	Afternoon	267	0.001059	6.818
●	Monday	0	-	-
	Tuesday	136	0.000941	7.112
	Wednesday	134	0.002121	6.212
	Thursday	131	0.001335	6.364
	Friday	0	-	-
	Saturday	0	-	-
◇	June	104	0.001854	6.526
	August	107	0.00091	6.843
	September	102	0.001416	6.777
	December	88	0.001515	5.943
◆	Lateness of start time 0 or less	327	0.000986	6.688
	Lateness of start time bigger than 0	74	0.003026	6.449
□	Number of receipts on the day 0 or less	401	0.001353	6.604
	Number of receipts on the day 1 or more	0	-	-
■	Number of no-shows 4 or less	175	0.001632	6.766
	Number of no-shows 5 or more	226	0.001191	6.339
☆	Proportion of first-timers less than 0.1	167	0.001706	6.212
	Proportion of first-timers more than 0.1	234	0.001206	6.819

Metamodeling equation: $y=ae^{bx}$.

**Table 4 ijerph-19-02073-t004:** Single-parameter optimization results for physician C-007.

Effective Parameters		Number of Patients	Weighting of Number of Patients	Improved Probability of Over-Waiting
○	Morning	134	0.9979	0.9985
	Afternoon	267	1.0007	
●	Monday	0	1.0000	0.9792
	Tuesday	136	0.9638	
	Wednesday	134	1.0035	
	Thursday	131	1.0333	
	Friday	0	1.0000	
	Saturday	0	1.0000	
◇	June	104	0.9489	0.9554 (3rd priority)
	August	107	1.0401	
	September	102	0.9628	
	December	88	1.0537	
◆	Lateness of start time 0 or less	327	1.0223	0.9347 (1st priority)
	Lateness of start time bigger than 0	74	0.9000	
□	Number of receipts on the day 0 or less	401	-	-
	Number of receipts on the day 1 or more	0	-	
■	Number of no-shows 4 or less	175	0.9028	0.9443 (2nd priority)
	Number of no-shows 5 or more	226	1.0748	
☆	Proportion of first-timers less than 0.1	167	1.0338	0.9848
	Proportion of first-timers more than 0.1	234	0.9755	

**Table 5 ijerph-19-02073-t005:** Single-parameter optimization results for physician B-006.

Effective Parameters		Number of Patients	Weighting of Number of Patients	Improved Probability of Over-Waiting
○	Morning	558	0.9973	0.3646
	Afternoon	14	1.0998	
●	Monday	213	1.0158	0.3615 (3rd priority)
	Tuesday	0	1.0000	
	Wednesday	236	0.9623	
	Thursday	0	1.0000	
	Friday	0	1.0000	
	Saturday	123	1.0441	
◇	June	127	1.0850	0.3372 (1st priority)
	August	121	0.9502	
	September	162	0.9856	
	December	162	0.9851	
◆	Lateness of start time 0 or less	254	1.0000	-
	Lateness of start time bigger than 0	318	1.0000	
□	Number of receipts on the day 0 or less	310	1.0000	-
	Number of receipts on the day 1 or more	262	1.0000	
■	Number of no-shows 9 or less	263	0.9542	0.3549 (2nd priority)
	Number of no-shows 10 or more	309	1.0386	
☆	Proportion of first-timers less than 0.3	319	1.0000	-
	Proportion of first-timers more than 0.3	253	1.0000	

**Table 6 ijerph-19-02073-t006:** Multi-parameter optimization results for each of the 21 physicians.

Physician ID Code	Normalized Probability of Over-Waiting			Three Most Effective Parameters Obtained from the Optimization		
	Initial	Optimal	Improvement	#1	#2	#3
C-007	1.00	0.82	0.18	◆	■	◇
A-009	0.91	0.77	0.14	■	●	□
C-004	0.84	0.77	0.07	○	○	◇
C-003	0.76	0.60	0.16	●	◇	○
C-001	0.57	0.26	0.31	◆	●	○
C-002	0.49	0.44	0.05	◇	●	○
C-018	0.43	0.13	0.30	●	◇	■
A-008	0.41	0.15	0.26	◇	●	■
A-014	0.38	0.18	0.20	●	■	◇
B-006	0.37	0.31	0.06	◇	■	◆
C-013	0.38	0.06	0.32	☆	●	◇
B-019	0.34	0.19	0.15	◇	□	●
B-017	0.30	0.12	0.18	◇	■	○
B-008	0.29	0.08	0.21	◇	☆	●
B-003	0.27	0.13	0.14	●	■	◇
A-006	0.26	0.06	0.20	●	◇	□
B-010	0.24	0.13	0.11	●	○	□
C-014	0.23	0.08	0.15	◇	●	○
A-015	0.17	0.12	0.05	●	◇	☆
A-001	0.15	0.05	0.10	●	□	◇
C-012	0.14	0.05	0.09	○	■	●

○: the session time of day, ●: day of the week, ◇: session month, ◆: start time delay, □: number of walk-in patients, ■: number of no-shows, ☆: proportion of first-time patients.

**Table 7 ijerph-19-02073-t007:** Multi-parameter optimization results for all 21 physicians.

Department	Normalized Probability of Over-Waiting		
	Prior	Improvement	Difference
All of A, B, and C	0.232	0.162	0.070
A	0.154	0.107	0.047
B	0.170	0.099	0.071
C	0.513	0.387	0.126

## Data Availability

Not applicable.

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
