# Peer review of "Physician-Customized Strategies for Reducing Outpatient Waiting Time in South Korea Using Queueing Theory and Probabilistic Metamodels"

_ijerph, 2022, doi:10.3390/ijerph19042073_

Round 1
Reviewer 1 Report
This study evaluates the parameters affecting long waiting time in outpatients’ departments optimizing schedules of these departments using queuing theory.
I found the article very interesting and useful to improve the waiting time in ambulatory care services. I only have some lower priority comments which I detail below. I wish the authors all the best taking his work forward.
- On page 5, line 200. It seems to me that it refers to the total number of patients in each session instead of the total number of no-shows.
- On page 6, table 1. Multiple regression analysis are performed but I think a better explanation of the methods used would be necessary.
- on page 7. To optimize the over-waiting probability of the patient for each physician you use an exponential function. It is not explained why you are using this function and not another. Is there bibliography on this?
- On page 8, figure 3. In the graph of figure 3, the y-axis represents the probability of over-waiting, but it is higher than 1!
- on page 10. In the paragraph of validation of effective parameters, it is said that the goodness-of-fit of the statistical model is very low, and consequently, the analysis is conducted for individual physicians and not for each department, but then, you don’t show the new results. In addition, a low R2 also indicates that more variables should be added, as the ones introduced explain little of the variability of the dependent variable.
Reviewer 2 Report
See some minor comments below

Reviewer 3 Report
Overall, a very interesting paper and a good review of how waiting times are currently being resolved. The authors have demonstrated the value of modelling in healthcare, particularly in the OPD, an area that is consistently struggling globally with wait times and physician issues. I have provided some specific comments below.
The Introduction does not flow well currently or provide a very good background. The authors jump between themes and ideas without good linkages. For example, what do you mean by access deficiencies? How do you define "providers" as opposed to staff? You have introduced the scheduling issue in Line 46 in terms of cost benefit but what is scheduling? It would be great to have it defined and linked to the other issues you have identified.
The first sentence doesn't read well - why is OP care the most important service the hospitals provide? rewording this would be beneficial. e.g. The most important factor in outpatient care is an affordable service.... etc.
Line 35 - waiting time can also refer to waiting for a nurse or nurse practitioner or an allied health professional? I would rephrase this to say "attended to by a healthcare professional", in OP, this could be any service.
Line 54 - missing "been" aimed at....
Line 49 - What do you mean by "provider supply"? Again, the terms are not well defined.
Line 56 - estimated? or evaluated?
Line 61 - What do you mean by "evidence-based parameters"? Most hospitals collect data around waiting times, staffing etc...
Lines 64 - 73 - A lot of re-working of language is required.
Figure 1 is not useful and is confusing. Consider removing it. The title of the figure does not make sense also.
Methods
This section could be revised in terms of what belong in methods and what belongs in results. I found it to be a little confusing at times as I thought I was reading the findings. Also, some language issues here. The statistical methods have been well described and appropriately align with the model that is being used.
I feel that some of the methods section actually belongs in the results - It would be beneficial to re assess this. For example, most of the tables and graphs display results.
Line 117 - what do you mean by "samples"? Is this the data?
Lines 127 - 129 - should be in the results section.
Line 137 - What is the "stand-by theory"? This has not previously been defined.
Line 147 - How did you come up with the parameters that may affect wait time? Are these from a previous study? Please define this.
Line 157 - Customer? or patient?
Line 188 - ensure you keep the tense consistent (is vs was).
Line 244 - this belongs in the results.
Results
The results are fairly well presented. I find the tables easy to follow, although the language could be slightly improved.
Line 329 - avoid using the word "appear to" - this makes it sound as though you are unsure of your results.
Line 390 - why do you call one group "physicians" and the other "doctors"? It would also be good to further define why you chose the 21 physicians for optimization (in methods).
Discussion
Overall, the discussion and conclusions are quite well written. It would be good to see your results linked more closely to the specific Korean systems, around individual parameters perhaps?
Line 411 - customer or patient? It would be beneficial to be consistent throughout. Patient would be preferred as you are working within healthcare.
An opportunity for further research would be to look at the qualitative parameters that effect patient waiting times as well. It would be good to see this in your limitations.
Round 2
Reviewer 3 Report
Thank you for your revisions based on previous feedback. You have improved the quality and rigour of your paper. I found the introduction much easier to follow, the terms are well defined and your aims and discussion are well aligned. The methods section is much clearer and the results link well to the methods. The language is very much improved and helps with the readability overall. Well done.